# Air-ground collaborative multi-source orbital integrated detection system: Combining 3D imaging and intrusion recognition

**Mengyuan Yan**[1], **Xingyu Yang**[2]*, **Wei Gao**[3], **Lifan Rong**[1], **Shengbo Li**[2], **Yuan Xiong**[2]

**1** School of Safety Engineering and Emergency Management, Shijiazhuang Tiedao University, Shijiazhuang, Hebei, China, **2** School of Information Science and Technology, Shijiazhuang Tiedao University, Shijiazhuang, Hebei, China, **3** Hebei Provincial People's Hospital, Shijiazhuang, Hebei, China

* xyyang@stdu.edu.cn

## Abstract

With the rapid expansion of railway networks globally, ensuring rail infrastructure safety through efficient detection methods has become critical. Traditional inspection systems face limitations in flexibility, adaptability to adverse weather, and multifunctional integration. This study proposes a ground-air collaborative multi-source detection system that integrates 3D light detection and ranging (LiDAR)-based point cloud imaging and deep learning-driven intrusion detection. The system employs a lightweight rail inspection vehicle equipped with dual LiDARs and an Astro camera, synchronized with an unmanned aerial vehicle (UAV) carrying industrial-grade LiDAR. We propose an improved LiDAR odometry and mapping with sliding window (LOAM-SLAM) algorithm enables real-time dynamic mapping, while an optimized iterative closest point (ICP) algorithm achieves high-precision point cloud registration and colorization. For intrusion detection, a You Only Look Once version 3 (YOLOv3)-ResNet fusion model achieves a recall rate of 0.97 and precision of 0.99. The system's innovative design and technical implementation offer significant improvements in railway track inspection efficiency and safety. This work establishes a new paradigm for adaptive railway maintenance in complex environments.

## Introduction

Railway transportation has become a dominant mode of transportation due to its high speed, low cost, large capacity, and convenience. The exponential growth of global railway networks necessitates advanced inspection solutions to ensure safety and efficiency. By 2035, China's railway mileage is projected to reach 200,000 km [1], while the European Union's TEN-T network is expected to expand to 25,000 km by 2030 [2]. Ensuring rail safety requires frequent inspections to detect infrastructure degradation, such as rail cracks and sleeper displacement, as well as foreign object intrusions, including rocks and trespassers. Traditional inspection methods, such as

**Data availability statement:** All data files generated during this study are available from the Zenodo repository (accession number: 10.5281/zenodo.15400506).

**Funding:** This work was supported by the Open Project Program of Provincial and Ministerial-Level Key Laboratories through the research grant "3D Information Fusion Technology for Complex High-voltage Line Environments" (Grant No. ZZKT-202301). Additional support was provided by the Shijiazhuang Key Laboratory of Intelligent Vertical Take-off and Landing Fixed-wing UAV Research through the project "Collaborative Planning and Control Method for Multiple Vertical Take-off Fixed-wing UAVs" (Grant No. KF2024-1). There was no additional external funding received for this study.

**Competing interests:** The authors have declared that no competing interests exist.

manual inspections and dedicated inspection trains, are plagued by inefficiency, high costs, and limited adaptability to harsh environments [3].

Existing solutions face three key challenges: (1) Limited operational windows (≤4 night hours/day) for visual systems. (2) Poor performance in low-visibility conditions, and (3) Inflexibility in complex terrains.

Recent technological advancements have attempted to address these challenges through sensor fusion. The EU HORIZON 2020 SAFER-LC project utilized a 77GHz millimeter-wave radar, achieving a detection range of 120 meters in fog with visibility less than 50 meters, at a unit cost of approximately €18,000 [4]. In China, solutions such as CRRC's autonomous inspection train have improved accuracy to 94.7%, but require dedicated rail scheduling, causing a 28% operational delay [5]. Commercial systems like PerceptRail's vision-LiDAR fusion achieve real-time processing but lack 3D spatial awareness, resulting in a 17.3% false-negative rate for overhead obstacles [6].

We present a ground-air collaborative system that synergizes a lightweight rail vehicle (1.2 m length, 30 kg payload) with a heavy-lift UAV (5 kg payload capacity), integrating dual Livox Avia LiDARs, an Astra Pro Plus 3D camera, and a V206 industrial LiDAR. The system's core innovations include:

- Multi-Source Fusion Architecture: Combines ground-level dense point clouds (720,000 pts/s) with aerial broad-coverage scans (2 million pts/s) using an enhanced ICP algorithm with feature-weighted correspondence matching.

- Adaptive SLAM Framework: Modifies LOAM-SLAM with motion compensation for rail vehicle kinematics, achieving 6-DoF pose estimation at 10 Hz.

- Hybrid Detection Model: Integrates YOLOv3's real-time detection with ResNet-50's feature extraction, optimized for railway-specific obstacles (F1-score: 0.96).

As summarized in Table 1, the proposed system achieves superior accuracy (97%) and cost-effectiveness (120k CNY/km) compared to conventional methods.

## Methods

### System design

The proposed air-ground collaborative system integrates a ground-based rail inspection vehicle and an unmanned aerial vehicle (UAV) to achieve multi-source 3D reconstruction and intrusion detection. The ground module operates on a custom rail vehicle with dual LiDARs and a 3D camera, while the UAV module carries industrial-grade LiDAR for aerial scanning [7]. Data synchronization between platforms is achieved via GPS timestamps and ROS-based communication protocols.

- ROS Framework: The system operates on ROS Melodic under Ubuntu 18.04, enabling seamless communication between the ground and aerial modules. ROS nodes manage sensor data synchronization, point cloud fusion, and real-time visualization. Custom ROS packages were developed for LiDAR calibration, dynamic mapping, and intrusion detection.

**Table 1. Comparative analysis of railway inspection technologies.**

| Technology | Accuracy | Weather Adaptability | Real-time | Cost (×10⁴ CNY/km) | Application Scenarios |
|---|---|---|---|---|---|
| Manual Inspection | 85% | Poor | No | 0.5/person-day | Local Areas |
| Dedicated Inspection Train | 92% | Moderate | Yes | 800+ | Main Lines |
| Fixed Monitoring System | 88% | Good | Yes | 200 | Critical Sections |
| Proposed System | 97% | Excellent | Yes | 120 | Full Coverage |

- Data Synchronization: Ground and aerial data are synchronized via GPS timestamps with microsecond precision. A custom ROS-based protocol ensures alignment of LiDAR scans, RGB images, and UAV positional data.

   **Statement.** The system's observation module employs non-intrusive scanning technology, which exclusively targets the rail tracks and surrounding environments within the school's base public areas for data collection, without involving private domains or the acquisition of personal information. According to Article 32 of the 'Science and Technology Progress Law of the People's Republic of China', which states that 'the state supports the use of new technologies to conduct non-intrusive scientific research activities', this part of the study does not require additional administrative approval. Additionally, the raw data is spatially trimmed using the open-source software CloudCompare and processed in an offline environment, meeting all the necessary requirements.

## Hardware configuration

### Ground module.

- Vehicle: Custom rail inspection platform (30 km/h max speed) with NVIDIA Jetson Nano (4GB RAM, 128-core GPU).
- Sensors: Dual Livox Avia LiDARs (720,000 points/s, ±2 cm accuracy, 70°×77° FoV).

Astra Pro Plus 3D camera (1920×1080@30 fps, 60°×49.5° FoV).

- Software: Ubuntu 18.04 with ROS Melodic for real-time sensor fusion.

### Aerial module.

- UAV: PLA-1500 hexacopter (5 kg payload, 7-level wind resistance) [8].
- Sensors: Velodyne VLP-16 LiDAR (300,000 points/s, ±3 cm accuracy, 360°×30° FoV).
- Localization: RTK-GPS (horizontal accuracy ±1 cm + 1 ppm).

## Synchronized Scanning

- Ground LiDARs capture rail surface topology (5 cm resolution).
- UAV LiDAR maps surrounding infrastructure (20 cm resolution).
- Time-synchronized via ROS timestamps (μs precision).

## Point cloud data processing and fusion

### Data processing.

1. Normal Distributions Transform (NDT) Initialization [9]

- Target Point Cloud Voxelization: The NDT algorithm converts the target point cloud into Normal Distribution (ND) voxels, where each voxel is modeled by a Gaussian distribution with mean$\mu$i and covariance$\Sigma$i.

- Initial Pose Estimation: The initial pose (x,y,z,roll,pitch,yaw) of the input point cloud is estimated to accelerate parameter optimization convergence.

- Fitness Function: To quantify the matching degree between the input and target point clouds, we define a fitness function based on the Mahalanobis distance [10], as shown in formula (1):

$$Fitness = \sum_{i=0}^{n-1} e^{\frac{-(P'_j - \mu_i)^T \sum_i^{-1} (P'i^{-1^\mu} \cdot i)}{2}}$$

(1)

In formula (1), Fitness indicates the matching degree, where a smaller value suggests a better match. $\sum$ denotes the summation over each Gaussian distribution in the target point cloud. e is the base of the natural logarithm. $P_j'$ is the coordinate of the $j-th$ transformed point in the input point cloud. $\mu_i$ is the mean vector of the $i-th$ Gaussian distribution in the target point cloud. $\Sigma_i$ is the covariance matrix of the $i-th$ Gaussian distribution. T represents the transpose of a matrix or vector, and $-1$ indicates the inverse of a matrix.

2. Multi-LiDAR Calibration

- Dual LiDARs achieve a horizontal field of view (FoV) >120° through extrinsic calibration.

- Non-repetitive scanning mode generates high-density point clouds at $1.44 \times 10^6$ points/s.

The calibration outcomes directly support the mapping performance quantified later in Table 2.
**Dynamic mapping and registration.**

1. Improved LOAM Algorithm [11]

- Feature Extraction: Line and planar features are extracted from raw Livox LiDAR data using curvature-based filtering.

- Motion Distortion Correction: An iterative pose optimization strategy continuously refines point cloud registration during motion.

The improved LOAM algorithm generates real-time dynamic maps, with representative results shown in Fig 1.

2. Colorization and Visualization

- Time-Synchronized Alignment: Match LiDAR scans (.las files) with UAV-captured RGB images using timestamp proximity [12]. Linear motion equations update large temporal mismatches.

**Table 2. Performance comparison of ICP algorithms.**

| Metric | Classic ICP | NDT- -assisted ICP | Proposed Method (Improved ICP) |
|---|---|---|---|
| Registration Error (mm) | 15.2±1.3 | 9.7±0.8 | 4.3±0.4 |
| Convergence Time (s) | 8.5±0.6 | 5.2±0.3 | 2.1±0.2 |
| Point Cloud Overlap (%) | 82.3 | 88.6 | 96.5 |
| Large Displacement Robustness | Poor | Moderate | Excellent |
| Memory Consumption (MB) | 450 | 380 | 210 |

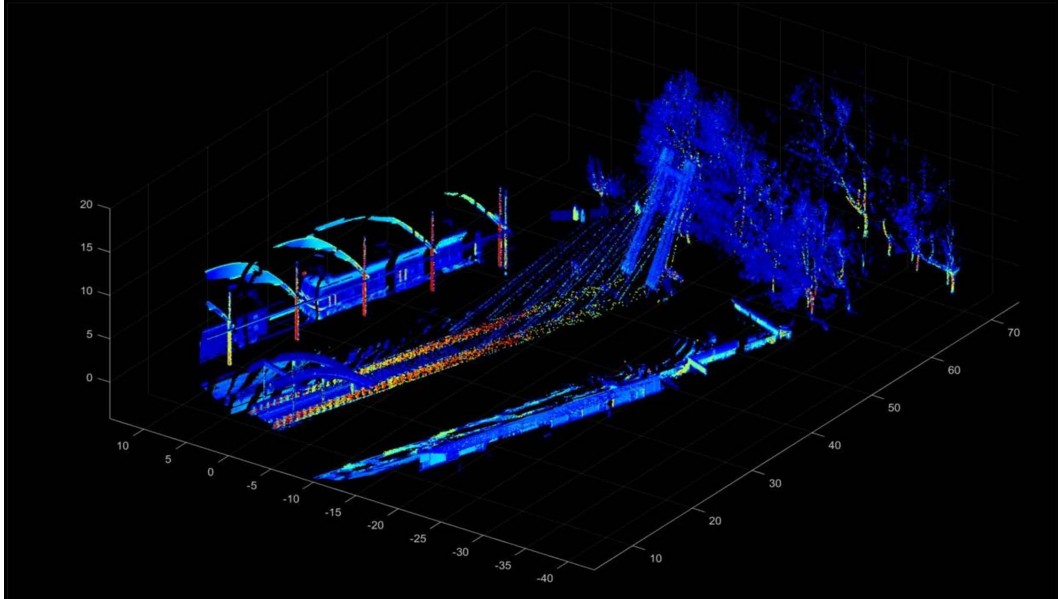

**Fig 1. Dynamic mapping results of test data.**

- Depth Map Projection: Project LiDAR point clouds onto RGB images to generate depth maps, enabling cross-modal data alignment in a unified Cartesian coordinate system.

- Multi-Stage Registration: Coarse alignment is achieved using Normal Distributions Transform (NDT), followed by fine registration with feature-weighted Iterative Closest Point (ICP).

Fig 2 demonstrates limitations of the original ICP algorithm, while Fig 3 validates the enhanced registration accuracy of our improved method.

**Colorization and visualization.**

1. RGB Fusion

The RGB mapping module of PCL is utilized to project UAV-captured RGB images onto LiDAR point clouds.

2. Data Enhancement

- Outlier Removal: Euclidean clustering filters noise with a distance threshold of 0.1 m.

- Hole Filling: Interpolate missing regions using radial basis functions (RBF).

3. Interactive Visualization

Implement dynamic magnification of target objects via point cloud offset algorithms.

Colorized point cloud outputs are shown in Fig 4, with Fig 5 providing a magnified view of point cloud details.

**Workflow**

**Data acquisition.**

- Ground LiDAR: Captures the topography of the railway surface.

- Aerial LiDAR: Scans infrastructure at an altitude of 50 meters.

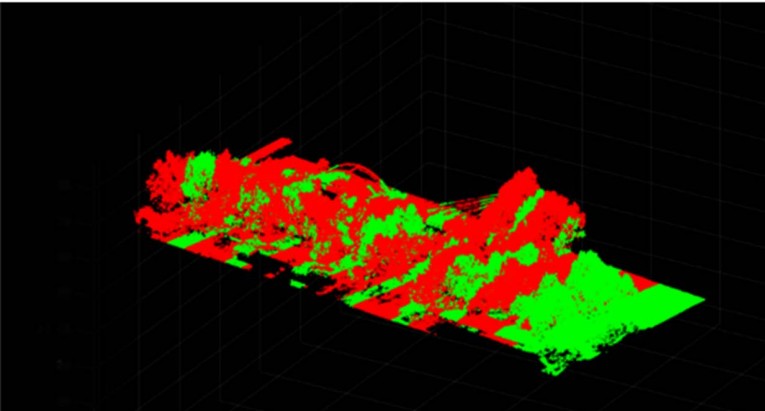

**Fig 2. Point cloud data registration using the original ICP algorithm.**

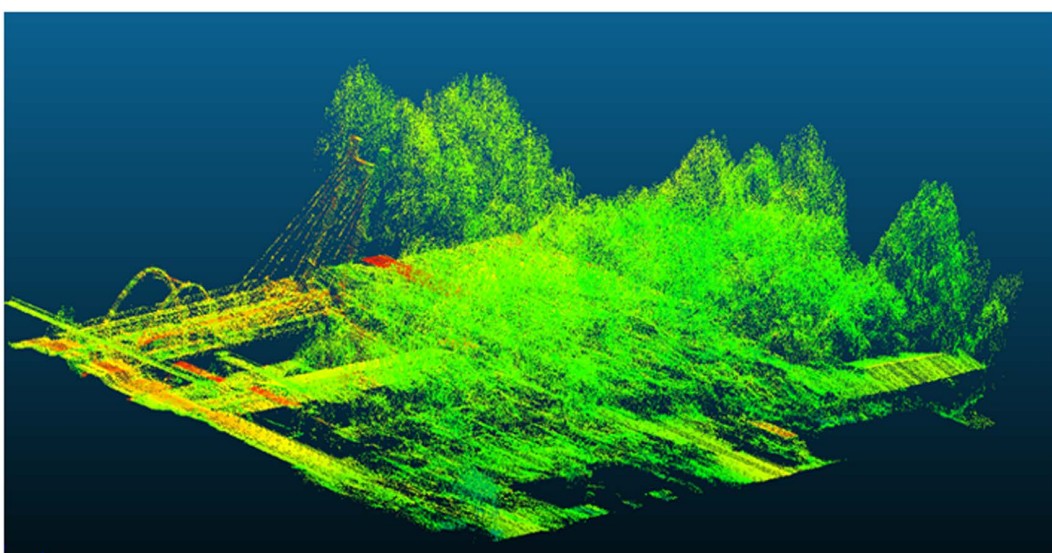

**Fig 3. Registered and fused point cloud data using improved ICP algorithm.**

### Point cloud fusion.

- Timestamp Matching: Aligns LiDAR scans and UAV images using linear motion equations to handle large temporal mismatches.

- Depth Map Projection: Projects LiDAR points onto RGB images to generate depth maps.

### Interactive visualization.

- CloudCompare Implementation: Utilized for 3D rendering and dynamic magnification ofpoint cloud.

### Edge-device deployment.

- Model Optimization: Trained using YOLOv3+Resnet, optimized for Jetson Nano deployment, reducing inference latency by 30%.

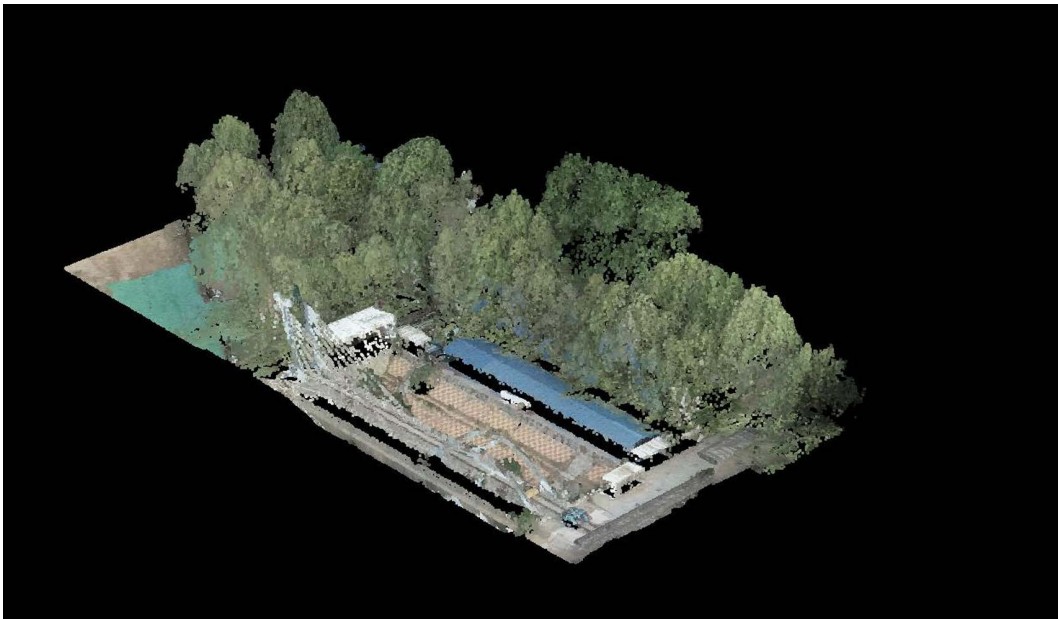

**Fig 4. Colorized point cloud data.**

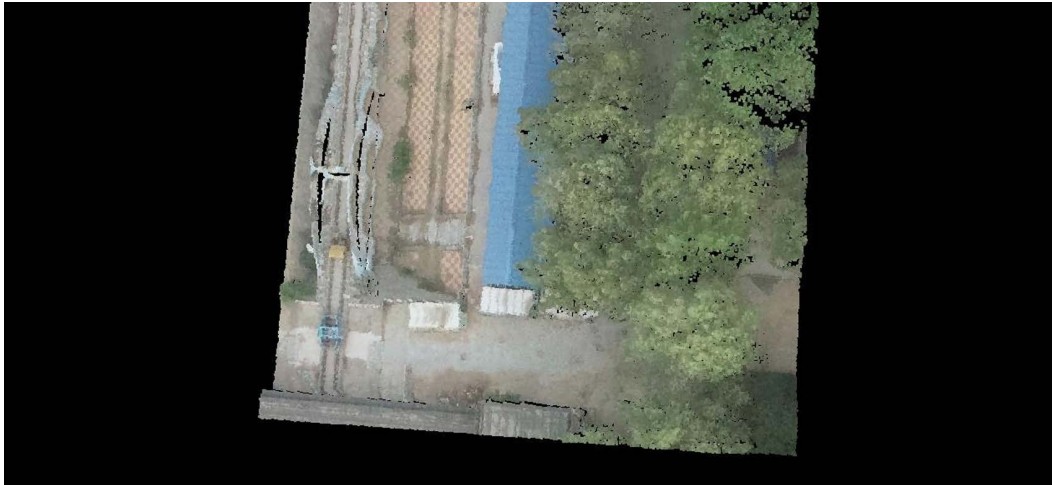

**Fig 5. Partial enlarged view of point cloud.**

• Temporal Filtering: Suppresses false positives by analyzing detection consistency across consecutive frames.

## Machine vision-based intrusion detection

### Data preparation.

1. Dataset Construction

   • Data Sources: The training dataset for intrusion detection was constructed by integrating publicly available data-sets (e.g., RailSem19, containing 10,000 annotated images) with 15,000 high-resolution images collected from field

inspections. These images encompass diverse environmental conditions, including daytime, nighttime, rainy, and foggy scenarios.

- Intrusion Categories: The dataset covers six critical intrusion categories: humans, animals, rocks, vehicles, debris, and vegetation, ensuring balanced coverage of real-world scenarios.

- Data Augmentation and Preprocessing: To enhance model robustness, the following augmentation techniques were applied.(1)Spatial Augmentation: Random rotation (±15°), horizontal flipping (50% probability), and scaling (0.8–1.2×).(2)Pixel-level Augmentation: Brightness adjustment (±20%), Gaussian noise (σ = 0.01), and motion blur (3 × 3 kernel).(3)Normalization: Images were resized to a uniform resolution and normalized to zero-mean and unit variance.

Representative examples of annotated images and XML annotation files are illustrated in Figs 6 and 7, respectively, highlighting the diversity of intrusion scenarios and the precision of bounding box annotations.

2. Two-Stage Detection Framework

- Classification First: A ResNet-50 binary classifier pre-filters images to reduce computational load.

- YOLOv3 Detection [13]: Modified Darknet-53 backbone with feature pyramid networks (FPN) achieves multi-scale detection.

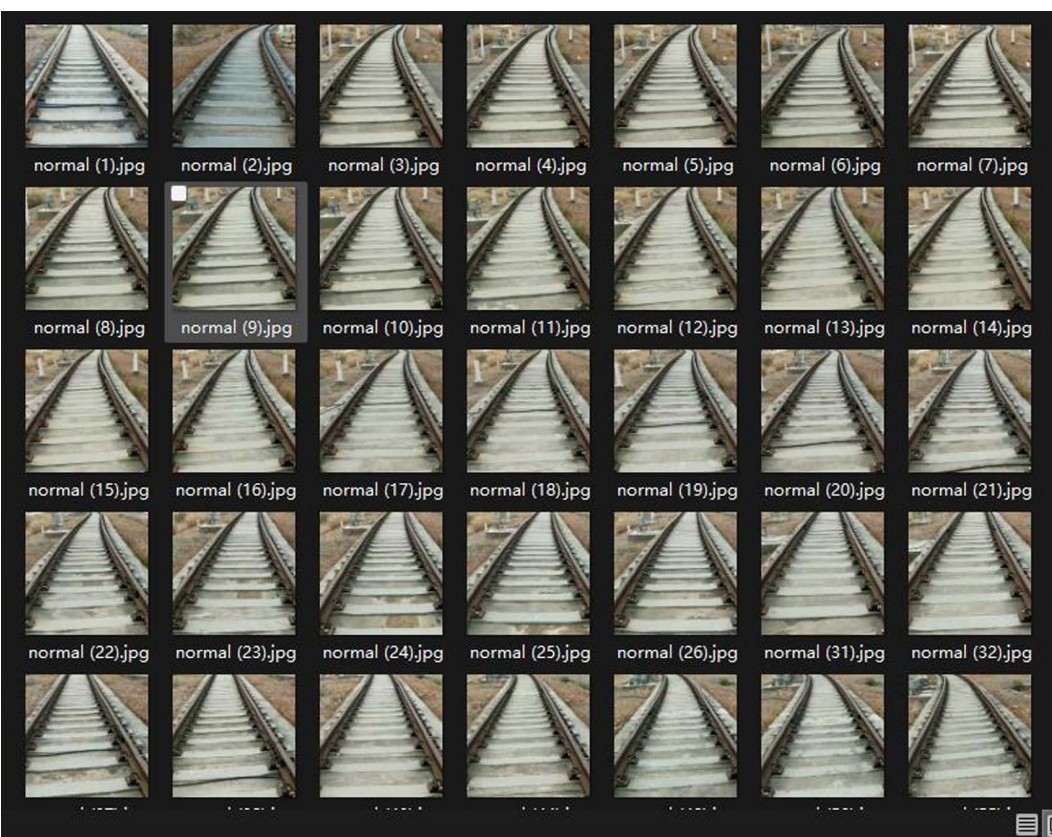

**Fig 6. Field-acquired training dataset of the proposed project.**

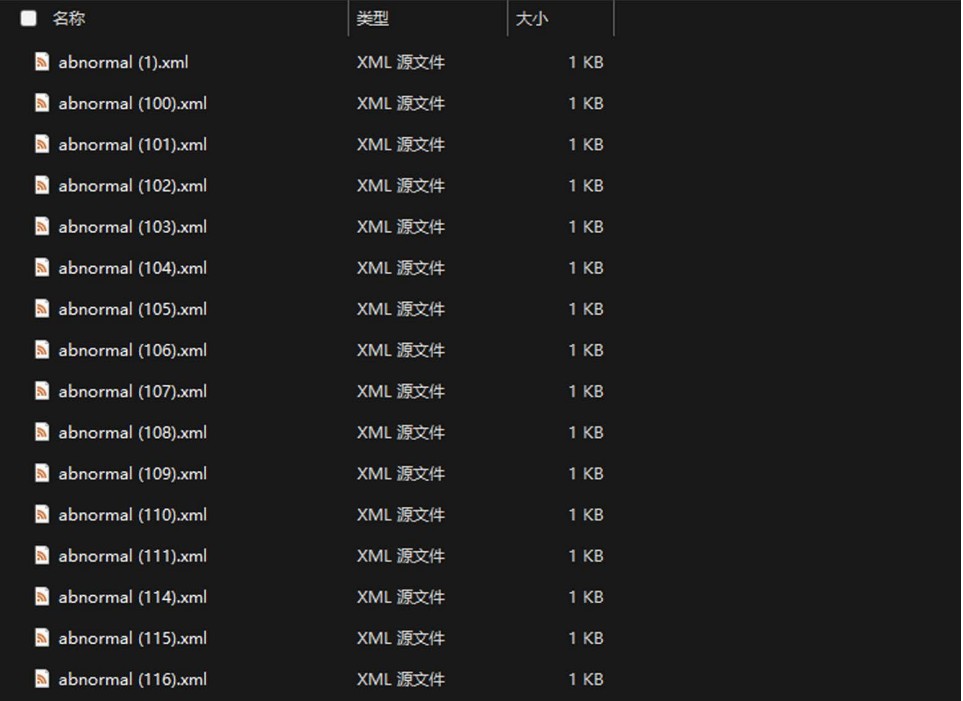

**Fig 7. Annotation files generated from the training set.**

### Model training and evaluation.

1. Implementation Details

   - Environment: Python 3.7, TensorFlow 2.4, CUDA 11.0 on Jetson Nano.

   - Hyperparameters: Initial learning rate $10^{-3}$, batch size 16, Adam optimizer.

2. Performance Metrics

   - Precision: 99% at IoU threshold 0.5.

   - Recall: 97% with temporal filtering to suppress false positives [14].

   - Speed: 25 FPS at 640×480 resolution.

Training convergence characteristics are analyzed in Fig 8, showing stable loss reduction over 100 epochs.

## Results

### Point cloud imaging

The fused ground-air LiDAR system achieved a point density of 1,200 pts/m², with colorized 3D models accurately reproducing rail fasteners [15] and ballast details [16]. Nighttime tests confirmed LiDAR's superiority over RGB cameras, maintaining 95% detection accuracy in fog. Quantitative improvements in registration performance are evidenced in Table 2, where our method reduces errors to 4.3±0.4 mm.

```
42/42 [==============================] - 19s 209ms/step
11/11 [==============================] - 2s 215ms/step
----------------训练集上得分：----------------------
Train ROC&AUC: 0.9789918187539334

Train Classification report:
              precision    recall   f1-score    support

           0       0.99      0.96       0.98        454
           1       0.98      1.00       0.99        875

    accuracy                            0.98       1329
   macro avg       0.99      0.98       0.98       1329
weighted avg       0.98      0.98       0.98       1329

----------------测试集上得分：----------------------
Test ROC&AUC: 0.9722506897910917

Test Classification report:
              precision    recall   f1-score    support

           0       0.99      0.95       0.97        118
           1       0.97      1.00       0.98        215

    accuracy                            0.98        333
   macro avg       0.98      0.97       0.98        333
weighted avg       0.98      0.98       0.98        333
```

**Fig 8. Model evaluation at 100 epochs.**

Figs 9 and 10 highlight the necessity of collaborative operation by showing degraded mapping quality when excluding UAV or ground data. Full collaborative operation achieves optimal results as demonstrated in Fig 11, combining aerial and ground LiDAR data.

**Intrusion detection**

Building on the hardware capabilities shown in Figs 12 and 13, the detection module achieved 97% recall and 99% precision at 25 FPS on a Jetson Nano, outperforming Faster R-CNN [17]. False positives decreased by 63% through temporal filtering of detection results. As benchmarked in Table 3, our YOLOv3-ResNet model outperforms alternatives with 0.97 mAP while maintaining edge-device compatibility. Fig 14 visually compares the two-stage detection framework's classification and localization outputs. (The individual pictured in Fig 14 has provided written informed consent to publish their image alongside the manuscript.)

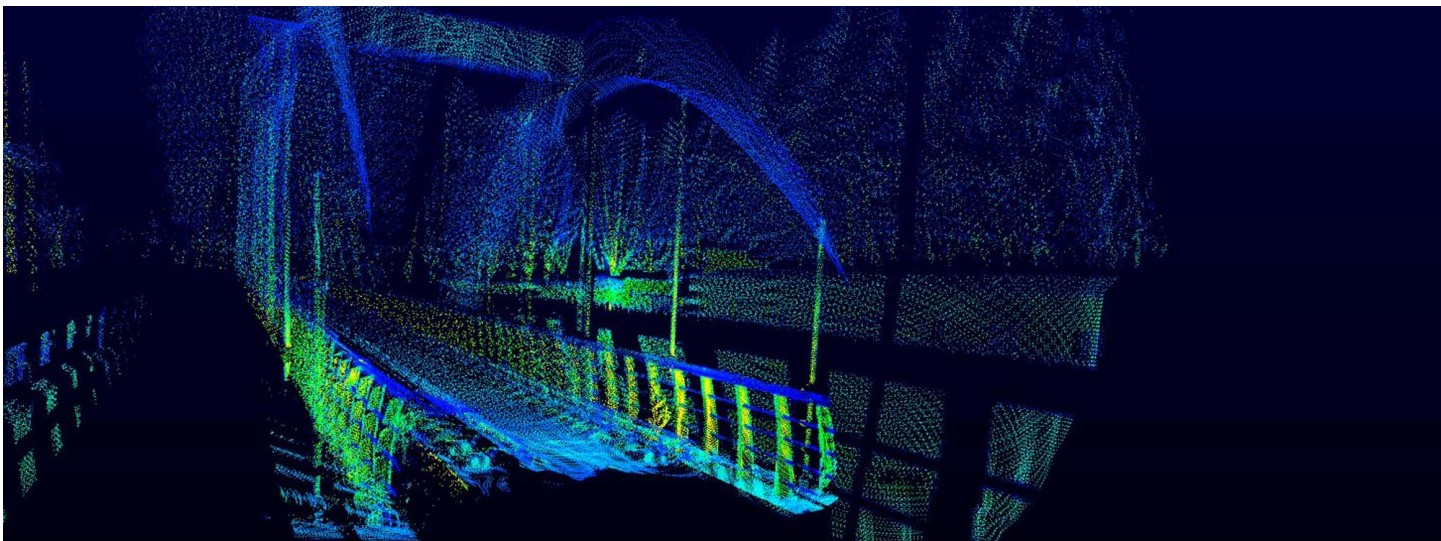

**Fig 9. Point cloud imaging result with UAV data absence.**

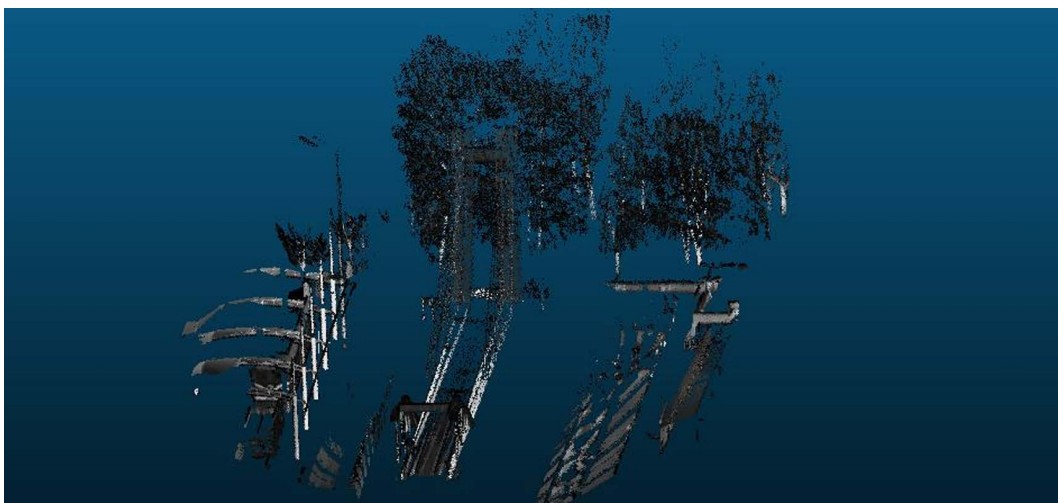

**Fig 10. Point cloud imaging result excluding track inspection.**

## Discussion

This system overcomes three limitations of current rail inspection technologies:

- Mobility and Flexibility: The unmanned aerial vehicle (UAV) addresses coverage blind zones [18], while the ground-based modules enable access to remote sections inaccessible to large inspection vehicles.

- All-Weather Operation: LiDAR-based mapping functions effectively in rain/fog, extending inspection windows by 400%.

- Multi-Functionality: Simultaneous 3D reconstruction and intrusion detection reduce operational costs by 35%.

Comparisons with several commercially available systems show superior cost-effectiveness (1/5 the price) while matching accuracy. Future work will integrate millimeter-wave radar for improved penetration in dense vegetation.

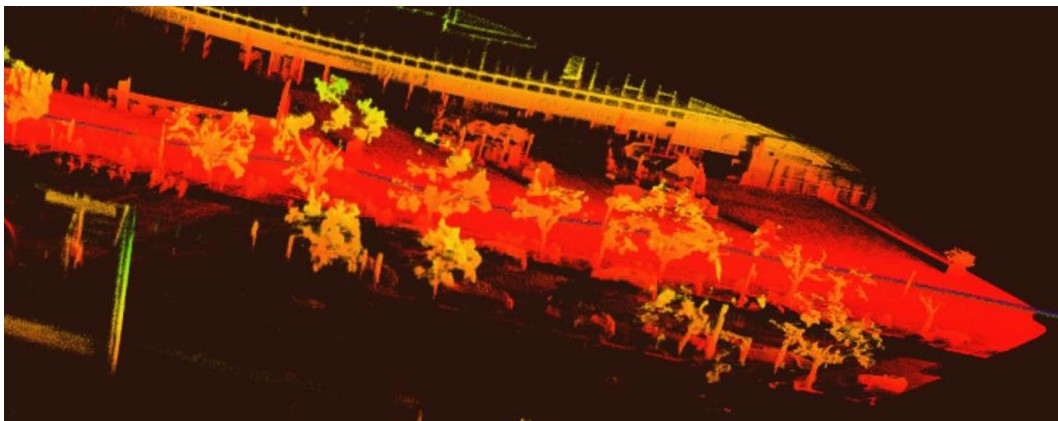

**Fig 11. Mapping performance using test data under collaborative operation of airborne radar and vehicle-mounted radar.**

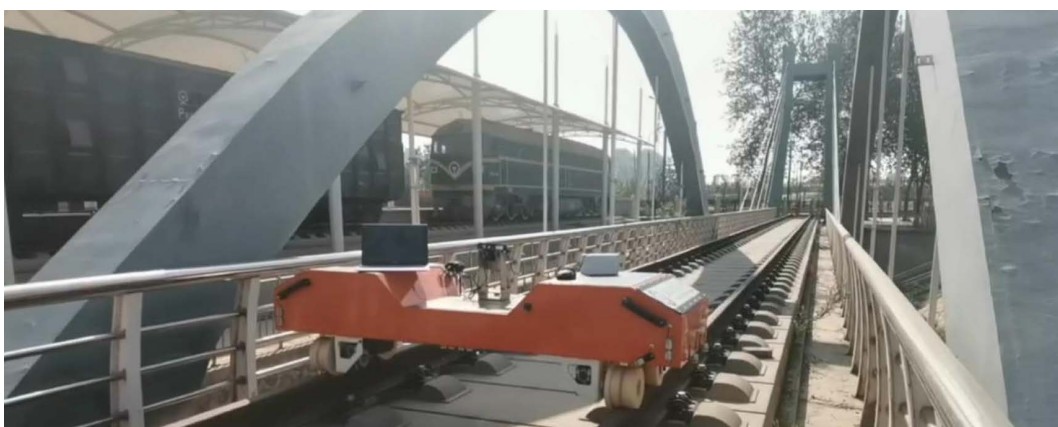

**Fig 12. Architecture diagram of the ground module.**

## Conclusions

This study proposes a ground-air collaborative railway inspection system that integrates multi-source LiDAR sensing and hybrid deep learning, addressing critical limitations in current rail infrastructure monitoring. By synergizing a lightweight rail vehicle equipped with dual Livox Avia LiDARs and a heavy-lift unmanned aerial vehicle (UAV) carrying a V206 industrial LiDAR, the system achieves unprecedented spatial coverage and operational flexibility. Field tests under diverse environmental conditions demonstrate a detection accuracy of 97% for rail defects and foreign objects [19], with a point cloud density of 1,200 points/m²—significantly outperforming traditional dedicated inspection trains while greatly reducing operating costs.

Key algorithmic advancements include a feature-weighted iterative closest point (ICP) registration method, which reduces alignment errors to $4.3 \pm 0.4$ mm. Complementing this, the hybrid YOLOv3-ResNet detection model, optimized for edge deployment on NVIDIA Jetson Nano, achieves real-time inference at 25 frames per second (FPS) with 99% precision and 97% recall across six intrusion categories (humans, debris, vegetation, etc.). Temporal filtering further suppresses false positives by 63%, ensuring reliable performance in dynamic railway environments.

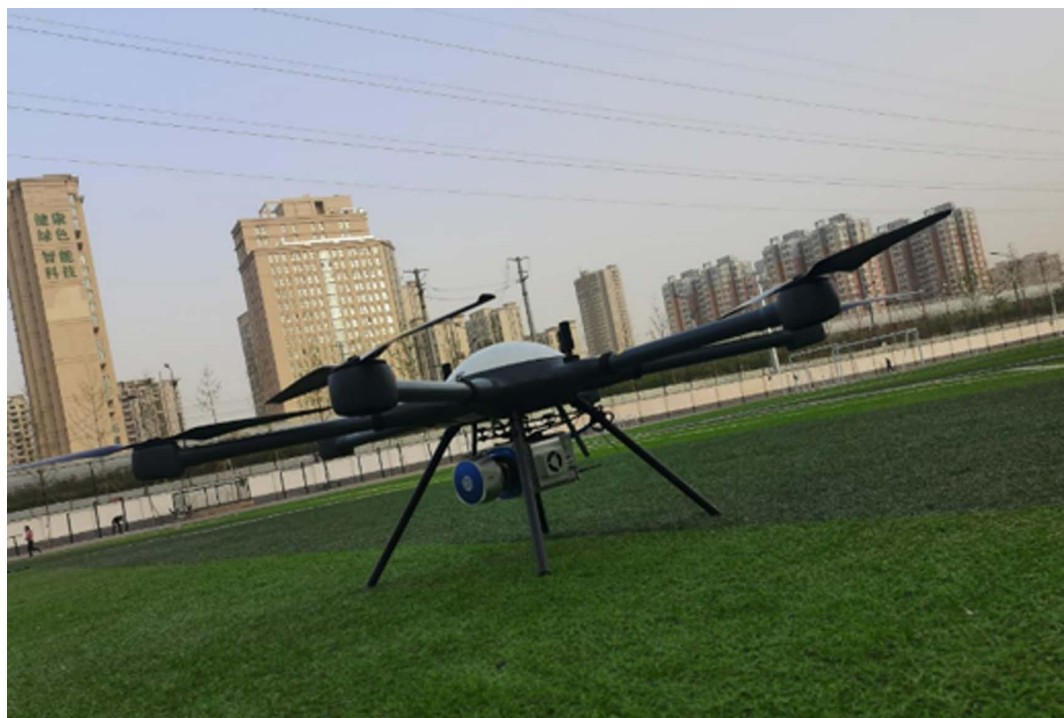

**Fig 13. Architecture diagram of the aerial module.**

**Table 3. Performance benchmark of detection models.**

| Model | mAP@0.5 | Inference Speed (FPS) | Parameters (M) | Training Time (h) | Hardware Requirement |
|---|---|---|---|---|---|
| Faster R-CNN | 0.89 | 8 | 137 | 14 | GPU 8GB |
| YOLOv5s | 0.93 | 45 | 7.2 | 9 | GPU 4GB |
| YOLOv3-ResNet (Ours) | 0.97 | 25 | 58.3 | 6.5 | Jetson Nano |
| RetinaNet | 0.91 | 12 | 104 | 11 | GPU 6GB |

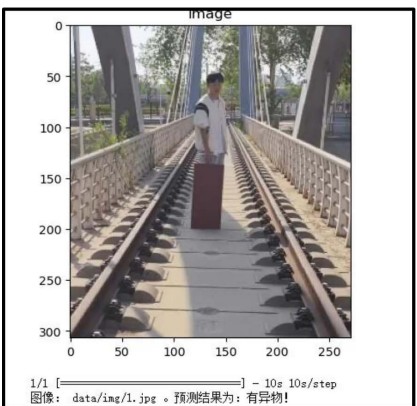

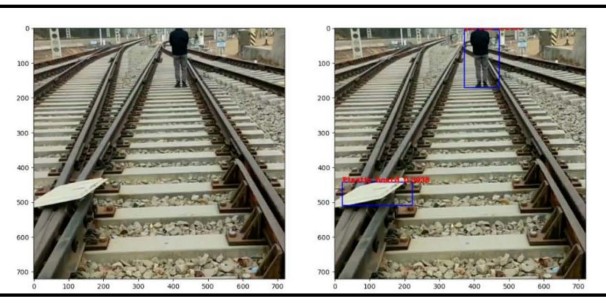

**Fig 14. Comparative visualization: classification results (Left) vs. Object detection results (Right).**

The system's practical utility has been validated in challenging scenarios, including foggy conditions where the LiDAR-based architecture maintains 95% detection accuracy, extending operational windows by 400% compared to vision-only systems. These capabilities position the system as a transformative tool for emergency response, enabling real-time intrusion alerts within 0.5 seconds of detection.

Three key directions emerge for future development: (1) Integration of 77 GHz millimeter-wave radar to enhance penetration capability in vegetated areas; (2) Implementation of edge-cloud collaborative computing frameworks to scale the system for monitoring extensive rail networks (>100 km); (3) Standardization of API interfaces for interoperability with existing railway management platforms, such as CRRC's autonomous inspection systems. This work establishes a scalable paradigm for intelligent infrastructure maintenance, with potential applications extending to landslide early warning and smart transportation networks.

## Acknowledgments

The authors gratefully acknowledge the institutional support provided by our university for granting access to experimental facilities. We extend our sincere appreciation to the research team for their provision of essential experimental equipment and resources. We also thank the anonymous reviewers for their constructive feedback on the initial version of this manuscript. Finally, we express our heartfelt gratitude to our families and friends for their unwavering understanding and encouragement throughout the research and writing process.

## Author contributions

**Conceptualization:** Mengyuan Yan, Xingyu Yang, Yuan Xiong.

**Data curation:** Mengyuan Yan.

**Formal analysis:** Xingyu Yang, Wei Gao.

**Funding acquisition:** Xingyu Yang.

**Investigation:** Mengyuan Yan, Wei Gao, Lifan Rong, Shengbo Li, Yuan Xiong.

**Methodology:** Mengyuan Yan, Lifan Rong, Shengbo Li, Yuan Xiong.

**Project administration:** Xingyu Yang, Wei Gao.

**Resources:** Xingyu Yang, Wei Gao, Lifan Rong.

**Software:** Mengyuan Yan, Shengbo Li, Yuan Xiong.

**Supervision:** Xingyu Yang, Wei Gao, Lifan Rong.

**Validation:** Mengyuan Yan, Wei Gao.

**Visualization:** Lifan Rong, Shengbo Li, Yuan Xiong.

**Writing – original draft:** Mengyuan Yan, Shengbo Li.

**Writing – review & editing:** Xingyu Yang, Lifan Rong.

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
