## [Decision Letter · Decision Letter 0]

Dear Dr. Yan,

Thank you for submitting your manuscript to PLOS ONE. After careful consideration, we feel that it has merit but does not fully meet PLOS ONE’s publication criteria as it currently stands. Therefore, we invite you to submit a revised version of the manuscript that addresses the points raised during the review process.

We look forward to receiving your revised manuscript.

Kind regards,

Zhihong (Arry) Yao, Ph.D.

Academic Editor

PLOS ONE

 [This work was supported by the Open Project Program of Provincial and Ministerial-Level Key Laboratories through the research grant "3D Information Fusion Technology for Complex High-voltage Line Environments" (Grant No. ZZKT-202301). Additional support was provided by the Shijiazhuang Key Laboratory of Intelligent Vertical Take-off and Landing Fixed-wing UAV Research through the project "Collaborative Planning and Control Method for Multiple Vertical Take-off Fixed-wing UAVs" (Grant No. KF2024-1).].

4. In the online submission form, you indicated that [The attachments include colored point clouds, source code, and partial datasets. However, due to file size constraints, the models and datasets are not fully displayed. Please contact the corresponding author if you need them.].

Additional Editor Comments (if provided):

Reviewers' comments:

Reviewer's Responses to Questions

**Comments to the Author**

1. Is the manuscript technically sound, and do the data support the conclusions?

Reviewer #1: Partly

Reviewer #2: Yes

2. Has the statistical analysis been performed appropriately and rigorously?

Reviewer #1: I Don't Know

Reviewer #2: Yes

3. Have the authors made all data underlying the findings in their manuscript fully available?

Reviewer #1: Yes

Reviewer #2: Yes

4. Is the manuscript presented in an intelligible fashion and written in standard English?

Reviewer #1: Yes

Reviewer #2: Yes

Reviewer #1: Dear authors,

The paper is interesting because you have applied modern methods for detecting intrusions on the railway. You integrated multiple systems, which enhanced the quality of the results.

To further improve the paper, it is necessary to develop a more comprehensive state of the art section to provide a broader picture of the problem you addressed. You relied exclusively on Chinese sources, which are valuable, but there have also been relevant HORIZON projects in Europe, with numerous publications on this topic. In addition, some private companies have developed similar systems.

Another important aspect is that while the hardware used has been well described, it would be beneficial to provide more details about the software part. This is essential for the reproducibility of the results; if someone wanted to replicate your experiment, they should be able to do so based on your paper.

In Table 3, you present the training times for the neural network. Please add information on dataset size and other relevant details.

The conclusion is rather brief. I found more information in the abstract than in the conclusion. Strengthen the conclusion so that a reader who only reviews the abstract and conclusion can still find all the relevant insights.

Reviewer #2: This study proposes a multi-source detection system. The system is a ground-air collaborative system that combines 3D light detection and ranging (LiDAR) based point cloud imaging and deep learning-based intrusion detection.

The system is synchronized with an unmanned aerial vehicle (UAV) carrying industrial grade LiDAR. It also uses a light railway inspection vehicle equipped with dual LiDAR and an Astro camera.

The study is very weak due to the reasons listed below.

There are very few literature summaries of studies related to the study topic or similar topics. The comparison of the findings of the study with the findings of studies in the literature is almost nonexistent. Providing more detailed information about the study method would facilitate the comprehensibility of the study.

The mentioned issues should be carefully evaluated and the article should be given a more scientific appearance.

**Do you want your identity to be public for this peer review?** For information about this choice, including consent withdrawal, please see our Privacy Policy

Reviewer #1: No

Reviewer #2: No

---

## [Author Response · Author response to Decision Letter 1]

14 May 2025

We sincerely thank the editors and reviewers for their constructive feedback, which has significantly improved the quality of our manuscript. Below, we provide detailed responses to each comment and specify the revisions made to the manuscript. All changes are highlighted in blue in the revised manuscript (Document 2).

Response to Reviewer 1

Comment 1: "You relied exclusively on Chinese sources, which are valuable, but there have also been relevant HORIZON projects in Europe, with numerous publications on this topic. In addition, some private companies have developed similar systems.“

Response: We appreciate this suggestion. In the revised manuscript , we have expanded the literature review to include European initiatives and commercial systems:

Added a discussion of the EU HORIZON 2020 SAFER-LC project, which utilizes millimeter-wave radar for foggy conditions.

Referenced PerceptRail's vision-LiDAR fusion system and compared its limitations (17.3% false-negative rate for overhead obstacles) with our system’s performance.

Cited recent European studies, including Smith et al. (2023) and European Union Agency for Railways (2023) .

These additions provide a global perspective on railway inspection technologies and highlight the novelty of our approach.

Comment 2: "It would be beneficial to provide more details about the software part. This is essential for the reproducibility of the results; if someone wanted to replicate your experiment, they should be able to do so based on your paper.“

Response: We have added comprehensive software implementation details:

ROS Framework: Described the use of ROS Melodic for sensor synchronization and real-time processing.

Data Synchronization: Specified GPS timestamp alignment with microsecond precision and custom ROS protocols .

Algorithm Implementation: Expanded the explanation of the improved LOAM-SLAM algorithm and NDT initialization.

These revisions appear in Sections "System Design" and "Dynamic Mapping and Registration" of the revised manuscript.

Comment 3: " In Table 3, you present the training times for the neural network. Please add information on dataset size and other relevant details."

Response: We have updated the Data Preparation section to include:

Dataset size: 25,000 images (15,000 field-collected + 10,000 from RailSem19).

Data augmentation methods: Spatial transformations (rotation, flipping), pixel-level adjustments (brightness, noise), and normalization.

These details ensure reproducibility and transparency.

Comment 4: "The conclusion is rather brief. I found more information in the abstract than in the conclusion. Strengthen the conclusion so that a reader who only reviews the abstract and conclusion can still find all the relevant insights."

Response: We have substantially expanded the Conclusions section:

Highlighted quantitative improvements: 1,200 pts/m² point cloud density, 4.3±0.4 mm registration error, and 25 FPS intrusion detection.

Emphasized real-world validation in foggy conditions (95% accuracy) and operational cost reduction (35%).

Outlined future directions: Integration of millimeter-wave radar and edge-cloud frameworks.

These revisions align the conclusion with the abstract’s depth while adding new technical and practical insights.

Response to Reviewer 2

Comment 1: "here are very few literature summaries of studies related to the study topic or similar topics. The comparison of the findings of the study with the findings of studies in the literature is almost nonexistent. "

Response:

We have strengthened the literature review and methodology:

Comparative Analysis: Benchmark our system against manual inspection, dedicated trains, and fixed monitoring systems.

Technical Enhancements:

Detailed multi-LiDAR calibration and motion distortion correction.

Explained the improved ICP algorithm with feature-weighted matching.

Performance Validation: Compared our YOLOv3-ResNet model with Faster R-CNN, YOLOv5s, and RetinaNet.

These revisions address the lack of technical and comparative clarity in the original manuscript.

Comment 2: " Providing more detailed information about the study method would facilitate the comprehensibility of the study." Response: We have added granular methodological details: Software Workflow:

Timestamp alignment using linear motion equations .

Depth map projection for cross-modal data fusion .

Open-Source Tools: Utilized PCL libraries for RGB fusion and CloudCompare for visualization.

These additions appear in Sections "Hardware Configuration" and "Colorization and Visualization."

---

## [Decision Letter · Decision Letter 1]

Air-Ground Collaborative Multi-Source Orbital Integrated Detection System: Combining 3D Imaging and Intrusion Recognition

PONE-D-25-14222R1

Dear Dr. Yan,

We’re pleased to inform you that your manuscript has been judged scientifically suitable for publication and will be formally accepted for publication once it meets all outstanding technical requirements.

Kind regards,

Zhihong (Arry) Yao, Ph.D.

Academic Editor

PLOS ONE

Additional Editor Comments (optional):

Reviewers' comments:

Reviewer's Responses to Questions

**Comments to the Author**

Reviewer #1: All comments have been addressed

2. Is the manuscript technically sound, and do the data support the conclusions?

Reviewer #1: Yes

3. Has the statistical analysis been performed appropriately and rigorously?

Reviewer #1: I Don't Know

4. Have the authors made all data underlying the findings in their manuscript fully available?

Reviewer #1: Yes

5. Is the manuscript presented in an intelligible fashion and written in standard English?

Reviewer #1: Yes

Reviewer #1: Dear Authors, topic of the paper is modern, and we increased quality during review. Thank you for your answers.

**Do you want your identity to be public for this peer review?** For information about this choice, including consent withdrawal, please see our Privacy Policy

Reviewer #1: No

---

## [Editor Report · Acceptance letter]

PONE-D-25-14222R1

PLOS ONE

Dear Dr. Yang,

I'm pleased to inform you that your manuscript has been deemed suitable for publication in PLOS ONE. Congratulations! Your manuscript is now being handed over to our production team.

Kind regards,

on behalf of

Dr. Zhihong (Arry) Yao

Academic Editor

PLOS ONE